# *TP53* Co-Mutation Status Association with Clinical Outcomes in Patients with *EGFR*-Mutant Non-Small Cell Lung Cancer

**DOI:** 10.3390/cancers14246127

**Published:** 2022-12-12

**Authors:** Xiuning Le, Cliff Molife, Mark S. Leusch, Maria Teresa Rizzo, Patrick M. Peterson, Nicola Caria, Yongmei Chen, Elena Gonzalez Gugel, Carla Visseren-Grul

**Affiliations:** 1MD Anderson Cancer Center, University of Texas, Houston, TX 77030, USA; 2Eli Lilly and Company, Lilly Corporate Center, Indianapolis, IN 46225, USA; 3Eli Lilly and Company, 3528 BJ Utrecht, The Netherlands

**Keywords:** advanced non-small cell lung cancer, *EGFR*, *EGFR*-TKI, *TP53*, overall survival, progression-free survival

## Abstract

**Simple Summary:**

There is conflicting evidence on the impact of TP53 co-mutations on survival in patients with EGFR-mutated advanced non-small cell lung cancer (aNSCLC). This observational study used data from a de-identified database to assess the role of TP53 co-mutation on survival in patients with EGFR-mutated aNSCLC (n = 356). We observed a significant decrease in survival outcomes (namely real-world progression-free survival [rwPFS] and overall survival [OS]) duration in the group of aNSCLC patients with presence of TP53 co-mutation compared to those with wild-type TP53 tumors. TP53 co-mutations were also associated with worse outcomes by subgroups, including type of EGFR-mutation and first-line treatment. Together, these findings indicated negative impact of TP53 co-mutation in outcomes for patients with EGFR-mutated aNSCLC.

**Abstract:**

*TP53* co-mutations have shown association with poor prognosis in various solid tumors. For *EGFR*-mutated advanced non-small cell lung cancer (aNSCLC), conflicting results exist regarding its impact on survival. Clinical outcomes and genomic data were obtained retrospectively from the real-world (rw) de-identified clinicogenomic database. Patients who initiated therapy for *EGFR*-mutated aNSCLC between January 2014 and December 2020 were identified. Clinical outcomes were evaluated by *TP53*-mutational status. In 356 eligible *EGFR*-mutated aNSCLC patients (median age 68 years), 210 (59.0%) had *TP53* co-mutation and 146 (41.0%) had *TP53* wild-type tumor. Unadjusted analysis showed significantly shorter survival in patients with *TP53* co-mutation versus *TP53* wild-type (rw progression-free survival [rwPFS]: HR = 1.4, 95% CI 1.1–1.9, *p* = 0.0196; overall survival [OS]: HR = 1.6, 95% CI 1.1–2.2, *p* = 0.0088). Multivariable analysis confirmed independent association between *TP53* co-mutation and worse rwPFS (HR = 1.4, 95% CI 1.0–0.9, *p* = 0.0280) and OS (HR = 1.4, 95% CI 1.0–2.0, *p* = 0.0345). Directionally consistent findings were observed for response rates, and subgroups by *EGFR*-activating mutation and first-line (1 L) therapy, with more pronounced negative effect in 1 L *EGFR*-TKI subgroup. *TP53* co-mutations negatively affected survival in patients with *EGFR*-mutated aNSCLC receiving standard 1 L therapy in real-world practice.

## 1. Introduction

Epidermal growth factor receptor (*EGFR*) mutations are among the most common oncogenic drivers in advanced or metastatic non-small cell lung cancer (aNSCLC) [1,2], which continues to be the most common cause of cancer deaths in the world [3]. Most (~85%) *EGFR*-activating mutations involve deletions in exon 19 (ex19del) or a substitution mutation in exon 21, specifically Leu858Arg (ex21L858R) [4] and they are classical *EGFR* mutations [5]. The current NCCN guideline recommendation and standard of care for aNSCLC with an activating *EGFR* mutation is targeted therapy with *EGFR*-Tyrosine Kinase Inhibitors (*EGFR*-TKIs) alone or in combination with antiangiogenics, ramucirumab or bevacizumab [4,6,7,8,9]. *EGFR*-TKIs are initially effective with progression-free survival ranging between 9 and 19 months [10,11,12,13,14,15]. However, resistance inevitably develops after initial benefit to *EGFR*-TKIs, leading to treatment failure. A few mechanisms of primary or acquired resistance have been identified, which include on-target *EGFR*-dependent mechanisms of resistance (e.g., exon 20 *T790M* and *C797S* mutations) and off-target *EGFR*-independent mechanisms (e.g., histologic transformation to small cell lung cancer, or acquisition of other driver alterations such as *MET*, *HER2*, p53, or *PI3KCA* amplification) [16,17,18,19,20,21,22,23,24,25]. However, more investigation is needed to understand and overcome resistance to *EGFR*-TKIs.

Various co-occurring genetic alterations in *EGFR*-mutated aNSCLC have been linked to poorer outcomes compared with *EGFR*-mutated aNSCLC without co-occurring alterations [26]. *TP53*, also known as the “guardian of the genome”, encodes the p53 transcription factor and plays a pivotal role in controlling and maintaining the overall integrity and stability of the genome [27]. It is the most frequently co-mutated gene in *EGFR*-mutated aNSCLC (up to 72%) [28], and dysfunction of the p53 protein due to mutations contributes to tumor development, progression, and metastasis [27]. Furthermore, the presence of mutations that disrupt the normal function of p53 predisposes patients to primary or acquired resistance to *EGFR*-TKI therapy [29].

Previous studies examining the impact of co-mutations in *EGFR*-mutated aNSCLC treated with *EGFR*-TKIs demonstrated that the presence of *TP53* co-mutation might be a marker of poor prognosis [28,30,31,32,33]. However, other studies found no association between the presence of *TP53* co-mutations and clinical outcomes [34,35,36,37,38]. The conflicting results were likely due to small cohort sizes, for example n = 37 in one study [34], or due to differences in statistical analyses (e.g., univariate vs. multivariable analyses accounting for known prognostic factors such as age, baseline metastases, and ECOG performance status) [35]. In early disease stage lung cancers (stage I–III), the role of *TP53* mutation is more controversial [36,37]. To provide a further body of evidence by building on previous work [30,31,32,33,34,35,36,37,38], this study examined the impact of *TP53* co-mutations on the treatment outcomes in a cohort with *EGFR*-mutated aNSCLC using a nationally representative electronic medical records (EMR)-derived database linked with next-generation sequencing data. Subgroup analyses are reported based on *EGFR* mutation subtype and *EGFR*-TKI generation.

## 2. Materials and Methods

### 2.1. Study Design and Patients

This retrospective observational study used the nationwide de-identified Flatiron Health Foundation Medicine (FH-FMI) aNSCLC clinicogenomic database to select adults (≥18 years) with aNSCLC harboring an *EGFR* mutation, with or without the *TP53* gene mutation, who initiated first-line (1 L) therapy between January 2014 and September 2020 in routine clinical practice. The *EGFR* mutations included were the exon 19 deletions or the exon 21L858R mutation, evaluated in comprehensive genomic profiling baitsets DX1, T7, T5a, or T4b. The patient data originated from approximately 280 cancer clinics (~800 sites of care). Patients with evidence of other primary tumors, other than non-melanoma skin cancer, and those with no confirmed EMR activity within 90 days of aNSCLC diagnosis, were excluded. Patients were followed from the date of initiation of 1 L therapy (index date) until death, loss to follow-up, or end of the database (December 2020), whichever occurred first. Eligible patients were divided into two groups based on the status of the *TP53* gene mutation: *TP53* wild-type and *TP53* mutant groups.

The development, structure, and validation of the linked, de-identified FH-FMI clinicogenomic database have been described previously [39]. Briefly, the database contains retrospective longitudinal clinical data derived from EMR data, comprising patient-level structured and unstructured data (e.g., demographics, tumor stage and histology, performance status, tumor progression, death), curated via technology-enabled abstraction and linked to genomic data derived from FMI comprehensive genomic profiling (CGP) tests (e.g., next generation sequencing) in the FH-FMI clinicogenomic database by de-identified, deterministic matching. The clinicogenomic database included 12,726 patients with chart-confirmed aNSCLC receiving care at Flatiron Health’s network of community (~75%) and academic (~25%) oncology practices [40].

The study was exempt from institutional review board (IRB) approval because it was a retrospective study and used de-identified data in compliance with the Health Insurance Portability and Accountability Act. Approval from WCG IRB for data collection for this real-world cohort was obtained prior to study conduct and included a waiver of informed consent.

### 2.2. End Points

The main end point was real-world progression-free survival (rwPFS, the time until the first documented tumor progression, death, or loss to follow-up), which was chosen because it measures treatment effects without the cofounding influence of subsequent therapies. Secondary end points included overall survival (OS, time until death or loss to follow-up), real-world overall response rate (rwORR, documented complete response or partial response), real-world disease control rate (rwDCR, documented complete response or partial response or stable disease), and real-world duration of response (rwDoR: time from initial tumor response to disease progression or death, whichever occurred first). Real-world best overall response (rwBOR) was defined as best tumor response (from progressive disease, stable disease, partial response, to complete response) during treatment. All end points were indexed to the start of 1 L therapy unless otherwise noted.

### 2.3. Statistical Analysis

Baseline characteristics and time-to-event endpoints were analyzed and presented for all eligible patients, whereas analysis of tumor response end points was restricted to patients with post-baseline tumor response assessment data. Descriptive statistics, including frequencies and percentages for categorical variables and median and range for continuous variables, were used to summarize the demographic and clinical characteristics as well as treatments received. Differences between groups were assessed using tests appropriate for the distribution of each measure, including t-tests for continuous variables and chi-squared tests for categorical variables. The Kaplan–Meier method was used to estimate rwPFS and other time-to-event endpoints, along with median values and 95% confidence intervals (CIs). The difference between groups for time-to-event endpoints was assessed using the log-rank test.

To evaluate and identify the baseline characteristics potentially associated with outcome in the dataset, hazard ratios (HRs) for time-to-event endpoints and odds ratios (ORs) for response rates, along with the corresponding 95% CIs, were computed using univariate Cox-proportional hazards and logistic regression models, respectively. Independent variables for the univariate models (including age, sex, smoking history, Eastern Cooperative Oncology Group performance status [ECOG PS], presence of brain or bone metastases, *EGFR* mutation type, and *TP53* mutation status) were selected a priori as having a potential prognostic impact in *EGFR*-mutated NSCLC based on the literature [41,42,43,44,45,46,47]. Any factor found to be statistically significant for any endpoint, *p* < 0.05 in a univariate model, was then included as a covariate in all final multivariable Cox and logistic regression models, with one final multivariable model for each endpoint. The final multivariable models were the primary basis for investigating the independent prognostic effect of the *TP53* co-mutations. No data imputation was performed for missing values.

For rwPFS and rwDoR, patients who were alive and progression-free at the end of the dataset (December 2020) were censored at the date of their last clinic visit. For patients still alive at the end of the observation period, OS was censored at their last activity date in the EMR. Subgroup analyses were performed to assess the effect of *TP53* co-occurring mutations in subsets with *EGFR* exon 19 deletions, *EGFR* exon 21L858R mutations, and those treated with *EGFR*-TKI monotherapy, first-, second-, and third-generation *EGFR*-TKIs, and other non-*EGFR*-TKI therapy (including chemotherapy and immunotherapy). All statistical tests were two-sided, with a *p* value < 0.05 considered statistically significant. Statistical analyses were performed using SAS version 9.4 software (SAS Institute Inc., Cary, NC, USA).

## 3. Results

### 3.1. Baseline Demographics and Clinical Characteristics

A total of 356 patients with *EGFR*-mutated aNSCLC were eligible for the study (Figure 1), 210 (59.0%) of whom had a *TP53* co-mutation and 146 (41.0%) had *TP53* wild-type tumors. Baseline patient characteristics overall and by *TP53* co-mutation status are summarized in Table 1. Overall, the median age was 68 years (interquartile range [IQR] 60.5–76.1) and the majority were white (60.4%) and female (68.5%) with stage IV disease (86.2%) at initial diagnosis, no history of smoking (56.7%), and ECOG PS ≥ 1 (70.2%). The proportion of patients with a *TP53* co-mutation with exon 19 deletion or exon 21L858R mutation was 57.0% and 43.0%, respectively. Distribution of baseline characteristics was generally similar by *TP53* co-mutation status, although patients with a *TP53* co-mutation tended to be younger (<65 years: 43.3% vs. 31.5%) and white (62.3% vs. 57.5%).

### 3.2. Treatments Received

By inclusion criteria, all patients received 1 L systemic therapy (Table 2). However, 169 (47.5%) also received second-line (2 L) treatment during the observation period of the study. Treatment patterns were generally similar between the *TP53* co-mutation and *TP53* wild-type groups (Table 2). Overall, the majority received targeted therapy with *EGFR*-TKI monotherapy in 1 L (75.0%), with chemotherapy alone (10.0%) and a chemotherapy combination with immunotherapy (5.6%) as the next two common treatments administered in 1 L. In 2 L, *EGFR*-TKI monotherapy (67.5%) was most received treatment, with chemotherapy in combination with immunotherapy (7.3%), and chemotherapy alone (4.7%) as the second and third most common treatments administered in 2 L. Osimertinib was the most used *EGFR*-TKI in both 1 L and 2 L (43.3% and 47.9%, respectively), followed by erlotinib (18.3% and 8.9%), and afatinib (12.4% and 8.3%) (Appendix A).

### 3.3. Survival End Points

Through December 2020, the mean (SD) follow-up for all eligible patients was 19.9 (15.8) months (median: 15.9 months [IQR, 8.3–26.7]). In the overall population, the median rwPFS and median OS were 12.4 months (95% CI 10.6–14.1) and 28.7 months (95% CI 25.3–33.2) respectively. The rwPFS was significantly shorter in patients with *TP53* co-mutations than those with *TP53* wild-type tumors (median rwPFS 10.6 vs. 14.6 months, unadjusted HR = 1.4, 95% CI 1.1–1.9, *p* = 0.0196), as was OS (median OS 25.6 vs. 46.9 months, unadjusted HR = 1.6, 95% CI 1.1–2.2, *p* = 0.0088). One- and two-year survival probabilities were consistently lower for patients with *TP53* co-mutations (rwPFS: 47.0% and 18.0%; OS: 77.0% and 56.0%) than those with *TP53* wild-type tumors (rwPFS: 57.0% and 27.0%; OS: 77.0% and 65.0%) (Figure 2A,B).

Based on univariate analysis, age ≥ 75 years, male sex, brain metastases, bone metastases, ECOG PS ≥ 1, *EGFR* exon 21L858R mutation, smoking history, and the interaction between the *EGFR* mutation subtype and *TP53* co-mutation status were significantly associated with worse survival outcomes (rwPFS and OS, Table 3) in the overall population. After adjustment for these factors in multivariable analysis, the *TP53* co-mutations remained independently associated with a significantly greater risk for disease progression (rwPFS: HR = 1.4, 95% CI 1.0–1.9, *p* = 0.0280, Figure 2A) or death (OS: HR = 1.4, 95% CI 1.0–2.0, *p* = 0.0345, Figure 2B).

### 3.4. Tumor Response

Among the 275 patients with evaluable tumor response (77.2% of overall study cohort), the rwORR in the overall population was 78.0%. Univariate analysis (Appendix A) showed lower response rates in patients with a *TP53* co-mutation than in those with *TP53* wild-type tumors, although differences were not statistically significant (rwORR: 75.0% vs. 83.0% odds ratio (OR) = 0.6, 95% CI 0.3–1.1, *p* = 0.1262; rwDCR: 91.0% vs. 94.0%, OR = 0.6, 95% CI 0.2–1.6, *p* = 0.3324). A similar trend was observed for rwDoR (median 9.0 [95% CI 6.9–11.1] vs. 11.9 [95% CI 8.4–15.5] months), best response of CR (4.5% vs. 10.8%) and best response of progressive disease (8.9% vs. 5.4%), (Appendix A).

### 3.5. Subgroup Analysis

#### 3.5.1. EGFR Mutation Subtype

When analysis was restricted to the type of *EGFR*-activating mutation, the negative effect of *TP53* co-mutations on the rwPFS in the overall population (HR = 1.4, 95% CI 1.0–1.9, Figure 2) was directionally consistent across the subgroups by activating the *EGFR* mutation subtype, but was not statistically significant in the subgroups of patients with exon 19 deletion (13.2 months [95% CI 8.7–15.3] vs. 15.6 months [95% CI 11.2–19.9], HR = 1.5 [95% CI 1.1–2.2], *p* = 0.0536, Figure 3A) or those with exon 21L858R mutations (9.9 months [95% CI 7.3–12.3] vs. 14.6 months [95% CI 9.4–19.3], HR = 1.4 [95% CI 0.0–2.2], *p* = 0.1817, Figure 4A). Similarly, the negative effect of *TP53* co-mutations on OS in the overall population (HR = 1.4, 95% CI 1.0–2.0, *p* = 0.0345, Figure 2B) was directionally consistent with the *EGFR*-activating mutation subtype. However, the difference in OS was statistically significant only in the subgroup of patients with exon 19 deletions (median 25.6 [95% CI 21.7–29.7] vs. 47.1 [28.7-not achieved, NA], HR = 1.9, 95% CI 1.2–2.9, *p* = 0.0072, Figure 3B) and not significant in those with exon 21L858R mutations (median 25.3 [95% CI 17.7–35.4] vs. 28.7 [95% CI 21.5-NA], HR = 1.2, 95% CI 0.7–2.0, *p* = 0.4350), Figure 4B).

Appendix A presents the four-way comparison between the survival values for the included groups. Survival analysis for patients with *EGFR* exon 21L858R mutations and *TP53* co-mutations had the shortest median rwPFS (9.9 months) compared with 15.6 months, 14.6 months, and 13.2 months for patients with *TP53* wild-type plus *EGFR* exon 19 deletions, *TP53* wild-type plus *EGFR* exon 21L858R mutations, and *TP53* co-mutations plus *EGFR* exon 19 deletions, respectively (*p* = 0.0243, Appendix A). Similarly, patients with *EGFR* exon 21L858R and *TP53* co-mutations had the shortest median OS (25.3 months) compared with 47.1 months, 28.7 months, and 25.6 months for patients with *TP53* wild-type plus *EGFR* exon 19 deletions, *TP53* wild-type plus *EGFR* exon 21L858R mutations, and *TP53* co-mutations plus *EGFR* exon 19 deletions, respectively (*p* = 0.0388; Appendix A).

#### 3.5.2. EGFR-TKI Monotherapy

When analysis was focused on the subset of patients receiving targeted therapy with *EGFR*-TKI monotherapy (75.0% of overall study population), the median rwPFS was 14.0 months (95% CI 11.3–15.6), the median OS was 29.2 months (95% CI 25.5–36.6), and the rwORR was 77.0% (95% CI 72.0–83.0%). The rwPFS was significantly shorter for patients with *TP53* co-mutations than those with *TP53* wild-type tumors (median 12.0 vs. 16.4 months, HR = 1.6, 95% CI 1.1–2.3, *p* = 0.0069), as was OS (median 25.3 vs. 44.7 months, HR = 1.7, 95% CI 1.1–2.6, *p* = 0.0121). In contrast, there was no significant difference in the rwORR between the *TP53* co-mutation group and the *TP53* wild-type group (rwORR 73.0% vs. 84.0%, OR 0.53, 95% CI 0.27–1.03) (Appendix A). After adjustment for significant prognostic factors in multivariable analysis, *TP53* co-mutations remained significantly associated with a worse rwPFS (HR = 1.5, 95% CI 1.1–2.2, *p* = 0.0168) but not with OS (HR = 1.5, 95% CI 1.0–2.3, *p* = 0.0669).

#### 3.5.3. EGFR-TKI Generation

When patients were grouped according to generation of *EGFR*-TKIs, analyses showed generally similar trends to those observed in the overall population, i.e., shorter rwPFS and OS in patients with *TP53* co-mutations than in patients with *TP53* wild-type tumor (Appendix A). Specifically, the negative effect of a *TP53* co-mutation on survival outcomes was directionally consistent across the first- (median rwPFS: 9.0 vs. 11.3 months, HR = 1.9, 95% CI 1.0–3.5; median OS: 29.2 vs. 42.0 months, HR 1.6, 95% CI 0.9–3.0) and third-generation (median rwPFS: 17.0 vs. 18.4 months, HR = 1.3, 95% CI 0.8–2.2; median OS 31.2 vs. not reported [NR], HR 1.1, 95% CI 0.6–2.3) *EGFR*-TKI subgroups, but not in the second-generation *EGFR*-TKI subgroup (median rwPFS: 13.9 vs. 11.8 months, HR = 1.3, 95% CI 0.6–3.0; median OS 25.3 vs. NR months, HR = 3.1, 95% CI 1.2–8.4), (Appendix A). Except for the median rwPFS for first generation (*p* = 0.0399) and OS for second generation (*p* = 0.0236), the differences did not reach statistical significance, possibly owing to the small number of patients in these subgroups.

#### 3.5.4. Non-EGFR-TKI Monotherapy Treatment

Analysis of the subpopulation who received other treatments (i.e., non-TKI monotherapy, including chemotherapy alone or in combination with immunotherapy, as well as *EGFR*-TKI combinations) showed no significant differences between the *TP53* co-mutation and *TP53* wild-type groups in rwPFS (median 7.4 vs. 10.0 months, HR = 0.9, 95% CI 0.5–1.6), and OS (median 25.6 vs. 24.5 months, HR = 1.3, 95% CI 0.7–2.3), although the results are limited by the relatively small sample size (n = 89 patients, 50 with *TP53* co-mutation and 39 with *TP53* wild-type tumor), Appendix A.

## 4. Discussion

In this large retrospective clinicogenomic dataset, real-world progression-free and overall survival were significantly shorter among treated *EGFR*-mutated aNSCLC patients with a *TP53* co-mutation versus those with a *TP53* wild-type tumor, even after adjustment for known clinicopathologic prognostic factors. A non-significant trend toward lower tumor response rates was observed among patients with a *TP53* co-mutation. This negative contribution was generally independent of the type of 1 L treatment or *EGFR* mutation subtype. These findings confirm and strengthen available evidence from other datasets with limited follow-up [28,30,31,32,33,48,49], supporting the independent negative prognostic value of *TP53* co-mutation status. This may also help to identify a subset of patients with anticipated worse outcomes on 1 L aNSCLC treatment, particularly with *EGFR*-TKI monotherapy. While this study represents a large dataset, larger cohorts with *EGFR* and *TP53* co-mutated aNSCLC are needed to validate the current findings and to further clarify the contribution and underlying mechanism of a *TP53* co-mutation to primary or acquired resistance to *EGFR*-TKIs.

The study dataset was drawn predominantly from community oncology centers and restricted to patients with genomic testing. However, there was limited representation from academic practices where broad-based genomic sequencing may be part of research protocols, and from vulnerable patients with disproportionate access to physicians with distinct testing and practice patterns. The dataset is nationally illustrative of demographic (e.g., age, sex, and geographic location) and clinical (e.g., stage of disease) characteristics and distribution of patients across community and academic practices [50,51,52]. For example, the incidence of a *TP53* co-mutation in our study (59%) was within the range reported in other observational and clinical datasets (30–72%) of patients with *EGFR*-mutated aNSCLC, using various methods of *EGFR* detection with different sensitivity [28]. As also previously reported [31,48], *TP53* co-mutations were generally not associated with other measured prognostic factors (e.g., age, sex, smoking status, ECOG PS, baseline brain metastases), *EGFR* mutation subtype, or treatment assignment. Lastly, consistent with guideline recommendations [7,8] and prior studies [53], most patients with *EGFR*-mutated aNSCLC received *EGFR*-TKI monotherapy (predominantly osimertinib) in the 1 L and 2 L settings in current study, with few patients receiving 1 L *EGFR*-TKI combinations with either anti-angiogenics or chemotherapy.

In line with previous studies linking a *TP53* co-mutation with reduced sensitivity to *EGFR*-TKIs [29,49], a *TP53* co-mutation was associated with worse outcomes in the current dataset, even after adjustment for known aNSCLC prognostic factors including age, sex, smoking history, performance status, and baseline brain or bone metastases. Our findings also suggest that *TP53* co-mutations in the context of the *EGFR* mutation subtype may impact response to treatment and the survival outcomes, as the negative survival effect associated with *TP53* co-mutations has been observed to occur less among patients with *EGFR* exon 19 deletions than exon 21L858R mutations [31,32] and is likely because *EGFR* exon 19 deletions are known to be more responsive to *EGFR*-TKIs [53,54], as also demonstrated by our results. Notably, this finding appears to be driven by the robust OS in patients with *TP53* wild-type tumors plus the exon 19 deletion subgroup (47.1 months), as the OS was similar between the exon 19 deletion subgroup plus a *TP53* co-mutation (25.6 months) and an exon 21L858R mutation plus the *TP53* co-mutation subgroup (28.7 months). Sufficiently powered subgroup analyses on the effect of *TP53* co-mutations by the *EGFR* mutation subtype are warranted to validate these associations.

One of the strengths of this study is that it was conducted between 2014–2020, where the treatment landscape for *EGFR*-mutated NSCLC had dramatically changed, spanning the approval of multiple first-, second-, and third-generation *EGFR*-TKIs, and allowing for selection of a relatively large cohort with a long follow-up to assess the prognostic impact *TP53* co-mutations in this setting. Despite the large sample size and directional consistency in survival endpoints and response, as well as subgroup analysis, the relatively small numbers and limited follow-up among some subgroups may have (1) underestimated the number of patients (<50% in the current dataset) who ultimately did receive 2 L treatment, and (2) limited statistical power to detect significant differences. For instance, as FDA approval of osimertinib in the frontline setting was in April 2018 [55], findings from subgroup analysis by third-generation *EGFR*-TKIs had limited follow-up (less than 2 years) and should be validated in future studies as cohorts of patients treated with 1 L osimertinib mature in routine practice.

*TP53* co-mutation status did not appear to influence treatment selection of therapy for *EGFR*-mutated aNSCLC, likely because there are no approved therapies targeting *TP53* mutations. Although treatments targeting *TP53* co-mutations remain under investigation, our findings may have therapeutic implications given the expansion of precision oncology and increased adoption of next generation sequencing and multi-gene panels [7,8,56], resulting in increased availability of *TP53* status in this *EGFR*-mutated aNSCLC patient population. Specifically, these findings strengthen the rationale for clinicians to consider available or emerging *EGFR*-TKI combination strategies with other therapies such as antiangiogenics to delay resistance and prolong duration of targeted therapy for *EGFR*-mutated aNSCLC. Notably, in contrast to the body of evidence [28,30,31,32,33,48], including our findings, showing attenuated benefit with *EGFR*-TKI monotherapy in patients with *EGFR* and *TP53* co-mutated aNSCLC, a biomarker and subgroup analysis of the RELAY trial demonstrated that the addition of ramucirumab (anti-*VEGF* receptor inhibitor) to erlotinib (first-generation *EGFR*-TKI) was associated with prolonged PFS for *EGFR* and *TP53* co-mutated aNSCLC, both in patients with *EGFR* exon 19 deletions or exon 21L858R mutations [57,58]. *TP53* mutations have also been linked to VEGF expression across different solid tumor cancers therefore increasing the sensitivity of these tumors to anti-VEGF therapies [59]. While promising [57,60], however, the efficacy of *EGFR*-TKI combinations in *EGFR* and *TP53* co-mutated NSCLC is not established and needs further study in larger datasets. 

The limitations of this study are inherent in the nature of the analyzed data, collected during routine care mainly for clinical documentation, management, and billing, and not for research. The potential for bias exists from: (1) misclassification due to data entry and reporting errors in the EMR and use of database algorithms (e.g., line of therapy, adherence to oral *EGFR*-TKIs), (2) missing or incomplete data (e.g., ECOG performance status, comorbidities, tumor response to therapy), and (3) unmeasured or unknown confounding bias despite the use of multivariable regression to control for measured prognostic factors. However, there is no reason to believe that the validity of the dataset would be systematically biased according to *TP53* co-mutation status, although the association in this study should not imply causation. In addition, this study did not capture information on different types of *TP53* co-mutations and only *TP53* co-mutation status at baseline was considered. This is because the information on *TP53* mutation subtypes was not collected or available in the Flatiron database. As earlier reports suggest that mutations in *TP53* may induce resistance through different mechanisms in the cancer cell [61], further classification of distinct *TP53* mutation subtypes (e.g., disruptive vs. non-disruptive, missense vs. non-missense, at primary vs. metastatic lesions), both alone and in combination with other *EGFR* mutation subtypes (e.g., *EGFR* exon 20 *T790M*, the most common mechanism of resistance for 1st and 2nd generation *EGFR*-TKIs), may predict more precisely the impact of *TP53* co-mutation on prognosis and survival on *EGFR*-TKIs.

## 5. Conclusions

This large observational study confirms and strengthens the available evidence demonstrating that a *TP53* co-mutation identifies a subset of *EGFR*-mutated aNSCLC patients with worse outcomes on *EGFR*-TKI monotherapy as compared to those with *TP53* wild-type *EGFR*-mutated aNSCLC who might benefit from more intensified treatment schedules. Tailoring the treatment to individual patient characteristics (e.g., *EGFR* mutation subtype and co-occurring mutations) at baseline might be helpful to further improve treatment outcomes for patients with *EGFR*-mutated aNSCLC.

## Figures and Tables

**Figure 1 cancers-14-06127-f001:**
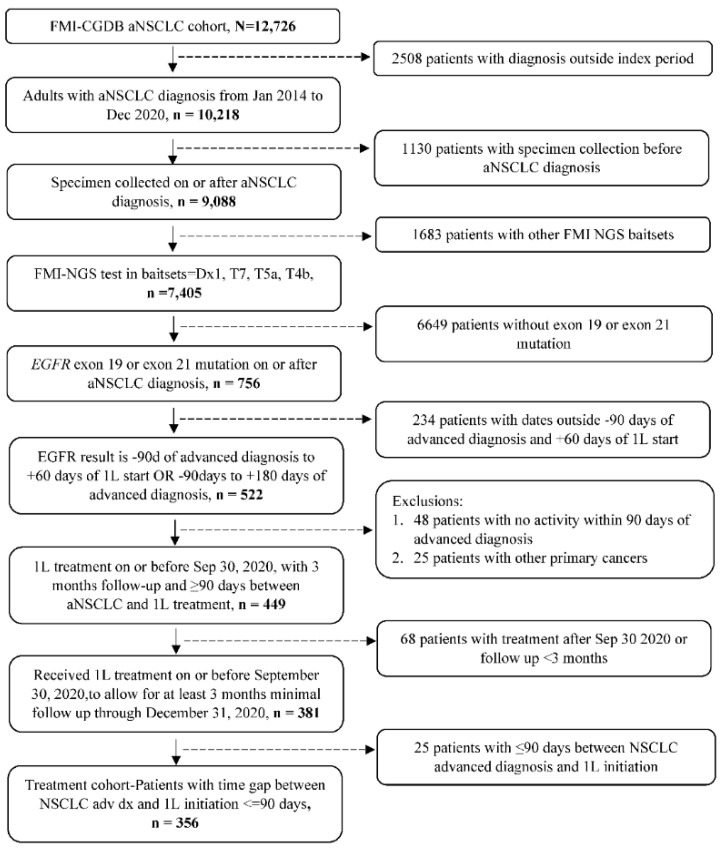
Patient inclusion/exclusion criteria and attrition.

**Figure 2 cancers-14-06127-f002:**
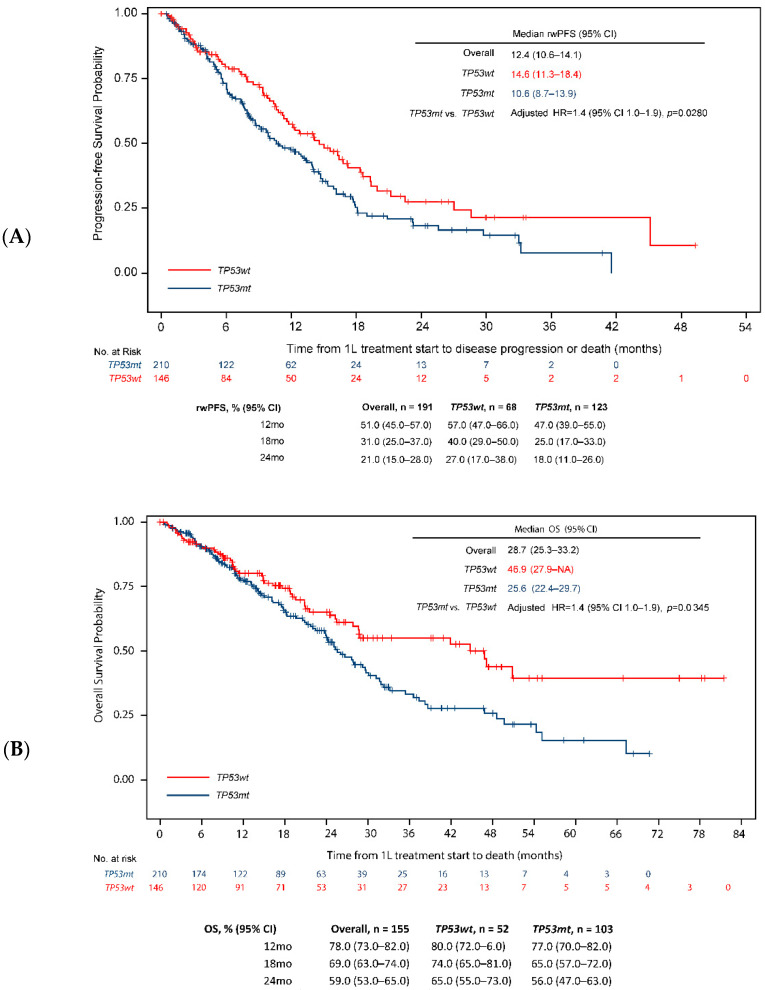
rwPFS and OS—*TP53* co-mutation vs. *TP53* wild-type. (**A**) rwPFS. (**B**) OS.CI = confidence interval; HR = hazard ratio; OS = overall survival; rwPFS = real-world progression-free survival; *TP53*mt = *TP53* co-mutation; *TP53*wt = *TP53* wild-type. *EGFR* mutation includes both exon 19 and exon 21 mutations.

**Figure 3 cancers-14-06127-f003:**
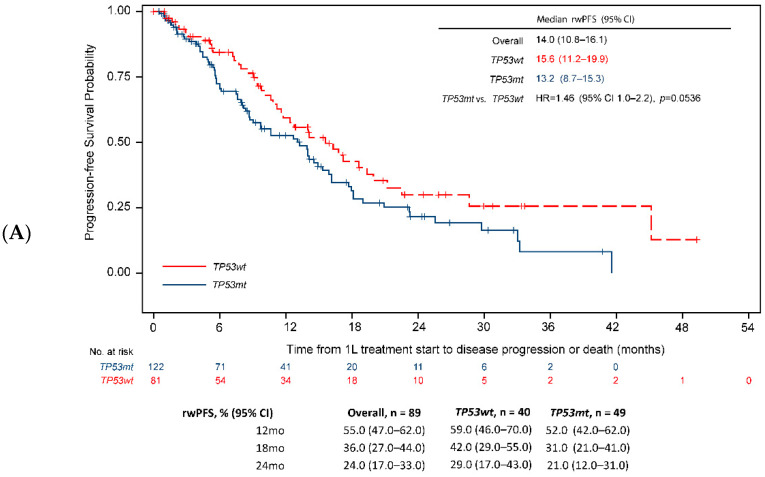
rwPFS and OS—*TP53* co-mutation vs. *TP53* wild-type by Exon19del mutation. (**A**) rwPFS. (**B**) OS. Unadjusted HR and 95% CI. CI = confidence interval; HR = hazard ratio; OS = overall survival; rwPFS = real-world progression-free survival; *TP53mt* = *TP53* co-mutation; *TP53wt* = *TP53* wild-type; exon 19 = exon 19 deletions.

**Figure 4 cancers-14-06127-f004:**
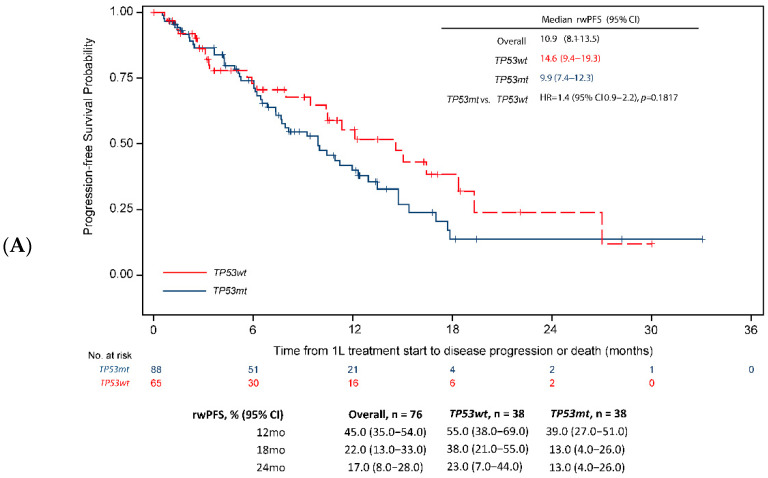
rwPFS and OS—*TP53* co-mutation vs. *TP53* wild-type by Exon21L858R mutation. (**A**) rwPFS. (**B**) OS. Unadjusted HR and 95% CI. CI = confidence interval; HR = hazard ratio; OS = overall survival; rwPFS = real-world progression-free survival; *TP53mt* = *TP53* co-mutation; *TP53wt* = *TP53* wild-type; exon 21 = exon 21L858R mutations.

**Table 1 cancers-14-06127-t001:** Patient demographic and clinical characteristics.

Variable	Overall (n = 356)	*TP53wt* (n = 146)	*TP53mt* (n = 210)
Follow-up in months from 1 L start, mean (SD)	19.9 (15.8)	21.5 (17.5)	18.8 (14.4)
Age at adv. Dx, median (IQR)	68.0 (60.5–76.1)	69.6 (63.5–77.7)	66.7 (58.2–75.2)
Age group, n (%)			
<65 years	137 (38.5)	46 (31.5)	91 (43.3)
65–74 years	115 (32.3)	51 (34.9)	64 (30.5)
75+ years	104 (29.2)	49 (33.6)	55 (26.2)
Sex, n (%)			
Female	244 (68.5)	103 (70.5)	141 (67.1)
Male	112 (31.5)	43 (29.5)	69 (32.9)
Race, n (%)			
Missing	26 (7.3)	9 (6.2)	17 (8.1)
Asian	41 (11.5)	22 (15.1)	19 (9.0)
Black or African American	15 (4.2)	5 (3.4)	10 (4.8)
Other Race	59 (16.6)	26 (17.8)	33 (15.7)
White	215 (60.4)	84 (57.5)	131 (62.4)
Smoking history, yes, n (%)	154 (43.3)	61 (41.8)	93 (44.3)
NSCLC stage at initial diagnosis, n (%)			
I–II	20 (5.6)	11 (7.5)	9 (4.3)
III	29 (8.1)	11 (7.5)	18 (8.6)
IV	307 (86.2)	124 (84.9)	183 (87.1)
ECOG performance status, n (%)			
0	107 (30.1)	43 (29.5)	64 (30.5)
1+	145 (40.7)	59 (40.4)	86 (41.0)
Missing	104 (29.2)	44 (30.1)	60 (28.6)
EGFR mutation type, n (%)			
Exon19	203 (57.0)	81 (55.5)	122 (58.1)
Exon21	153 (43.0)	65 (44.5)	88 (41.9)
Metastatic site, n (%)			
Brain	77 (21.6)	28 (19.2)	49 (23.3)
Bone	97 (27.2)	34 (23.3)	63 (30.0)
Liver	14 (3.9)	8 (5.5)	6 (2.9)
Lung	51 (14.3)	17 (11.6)	34 (16.2)
Lymph Node	23 (6.5)	13 (8.9)	10 (4.8)
Adrenal	21 (5.9)	7 (4.8)	14 (6.7)
Other	5 (1.4)	3 (2.1)	2 (1.0)
Practice type, n (%)			
Academic	41 (11.5)	18 (12.3)	23 (11.0)
Community	315 (88.5)	128 (87.7)	187 (89.0)

Adv. Dx = advanced diagnosis; ECOG = Eastern Cooperative Oncology Group; *EGFR* = epidermal growth factor receptor; IQR = interquartile range; NSCLC = non-small cell lung cancer; SD = standard deviation; *TP53* mt = *TP53* co-mutation; *TP53* wt = *TP53* wild-type. Smoking history includes both current and past smokers. *EGFR* mutation includes both exon 19 and exon 21 mutations.

**Table 2 cancers-14-06127-t002:** Treatment patterns in 1 L and 2 L—by drug class.

1 L				2 L			
Drug Class	Overall	*TP53wt*	*TP53mt*	Drug Class	Overall	*TP53wt*	*TP53mt*
	n = 356	n = 146	n = 210		n = 169	n = 61	n = 108
*EGFR*-TKI monotherapy	267 (75.0)	107 (73.6)	160 (76.2)	*EGFR*-TKI monotherapy	114 (67.5)	43 (70.5)	71 (70.3)
Chemotherapy alone	37 (10.0)	19 (13.1)	18 (8.6)	Chemotherapy + Immune Checkpoint Inhibitor	13 (7.3)	4 (6.6)	9 (7.4)
Chemotherapy + Immune Checkpoint Inhibitor	20 (5.6)	7 (4.3)	13 (6.2)	Chemotherapy alone	8 (4.7)	2 (2.9)	6 (5.9)
Other	8 (2.5)	3 (2.1)	5 (2.4)	Immune Checkpoint Inhibitor monotherapy	7 (4.1)	5 (8.2)	2 (1.9)
Chemotherapy + VEGF Inhibitor	8 (2.2)	3 (2.1)	5 (2.4)	*EGFR*-TKI + Chemotherapy + Immune Checkpoint Inhibitor	4 (2.4)	0 (0)	4 (3.9)
*EGFR*-TKI + VEGF Inhibitor	6 (1.7)	2 (1.4)	4 (1.9)	Other + Chemotherapy	3 (1.8)	2 (2.9)	1 (0.9)
*EGFR*-TKI + Chemotherapy	3 (0.9)	1 (0.7)	2 (0.9)	*EGFR*-TKI + Immune Checkpoint Inhibitor	3 (1.8)	0 (0)	3 (3.0)
Immune Checkpoint Inhibitor monotherapy	3 (0.9)	2 (1.4)	1 (0.5)	Chemotherapy + VEGF Inhibitor	3 (1.7)	0 (0)	3 (2.8)
*EGFR*-TKI + Chemotherapy + VEGF Inhibitor	1 (0.3)	0 (0)	1 (0.5)	*EGFR*-TKI + Chemotherapy	2 (1.2)	2 (3.3)	0 (0)
*EGFR*-TKI + Chemotherapy + Immune Checkpoint Inhibitor	1 (0.3)	1 (0.7)	0 (0)	*EGFR*-TKI + Chemotherapy + VEGF Inhibitor	2 (1.2)	0 (0)	2 (1.9)
*EGFR*-TKI + Immune Checkpoint Inhibitor	1 (0.3)	0 (0)	1 (0.5)	*EGFR*-TKI + VEGF Inhibitor	2 (1.2)	1 (1.64)	1 (0.9)
Other + *EGFR*-TKI	1 (0.3)	1 (0.7)	0 (0)	Hormone receptor therapy	1 (0.6)	0 (0)	1 (0.9)
				Immunosuppressant	1 (0.6)	0 (0)	1 (0.9)
				Other + *EGFR*-TKI	1 (0.6)	0 (0)	1 (0.9)

All data are presented as n (%). *EGFR*-TKI = epidermal growth factor receptor-tyrosine kinase inhibitor; Other = clinical study drug; *TP53mt* = *TP53* co-mutation; *TP53wt* = *TP53* wild-type; VEGF = vascular endothelial growth factor. *EGFR* mutation includes both exon 19 and exon 21 mutations.

**Table 3 cancers-14-06127-t003:** Univariate analysis—rwPFS and OS.

Variable	Category	rwPFS, HR (95% CI)	OS, HR (95% CI)
Overall			
Age at adv dx	≥75 vs. <75	1.2 (0.9–1.6)	2.0 (1.4–2.7) ^d^
Sex	Male vs. Female	1.1 (0.8–1.5)	1.4 (1.0–2.0) ^a^
ECOG PS	1+ vs. 0	1.5 (1.1–2.2) ^a^	1.7 (1.1–2.5) ^b^
	Unknown vs. 0	1.5 (1.0–2.2) ^a^	0.9 (0.6–1.5)
Smoking history	Yes vs. No	1.6 (1.2–2.1) ^b^	1.4 (0.99–1.87)
*TP53 status*	mt vs. wt	1.4 (1.1–1.9) ^a^	1.6 (1.1–2.2) ^b^
* TP53 status*/*EGFR* mutation type	*TP53mt*/Exon19 vs. *TP53wt*/Exon19	1.5 (1.0–2.2)	1.9 (1.2–2.9) ^b^
	*TP53mt*/Exon21 vs. *TP53wt*/Exon19	1.9 (1.3–2.9) ^b^	1.8 (1.1–3.0) ^a^
	*TP53wt*/Exon21 vs. *TP53wt*/Exon19	1.4 (0.9–2.3)	1.44 (0.8–2.5)
Bone mets	Yes vs. No	1.0 (0.7–1.4)	1.4 (1.0–1.9)
Brain mets	Yes vs. No	1.8 (1.3–2.5) ^c^	1.7 (1.2–2.4) ^b^
Exon group	Exon21 vs. Exon19	1.3 (1.0–1.8) ^a^	1.1 (0.8–1.5)
TKI Monotherapy			
Age at adv dx	≥75 vs. <75	1.1 (0.8–1.6)	1.7 (1.1–2.5) ^b^
Sex	Male vs. Female	1.2 (0.8–1.7)	1.5 (1.0–2.3) ^a^
ECOG PS	1+ vs. 0	2.1 (1.3–3.3) ^b^	2.4 (1.4–4.1) ^b^
	Unknown vs. 0	2.2 (1.4–3.5) ^b^	1.3 (0.7–2.3)
Smoking history	Yes vs. No	1.6 (1.2–2.2) ^b^	1.6 (1.1–2.3) ^a^
*TP53*	mt vs. wt	1.6 (1.1–2.3) ^b^	1.7 (1.1–2.6) ^a^
*TP53 status*/*EGFR* mutation type	*TP53mt*/Exon19 vs. *TP53wt*/Exon19	1.6 (1.0–2.4) ^a^	2.2 (1.3–3.7) ^b^
	*TP53mt*/Exon21 vs. *TP53wt*/Exon19	2.0 (1.2–3.2) ^b^	2.1 (1.1–3.9) ^a^
	*TP53wt*/Exon21 vs. *TP53wt*/Exon19	1.2 (0.6–2.1)	1.9 (0.9–3.7)
Bone mets	Yes vs. No	1.1 (0.8–1.6)	1.4 (0.9–2.1)
Brain mets	Yes vs. No	1.8 (1.2–2.6) ^b^	1.7 (1.1–2.6) ^b^
Exon group	Exon21 vs. Exon19	1.2 (0.9–1.7)	1.2 (0.8–1.8)
Other Monotherapy			
Age at adv dx	≥75 vs. <75	1.6 (0.8–3.0)	3.1 (1.7–5.6) ^c^
Sex	Male vs. Female	0.7 (0.4–1.4)	1.1 (0.6–2.1)
ECOG PS	1+ vs. 0	0.8 (0.4–1.6)	1.1 (0.6–2.0)
	Unknown vs. 0	0.6 (0.3–1.3)	0.5 (0.2–1.3)
Smoking history	Yes vs. No	1.4 (0.8–2.6)	1.0 (0.6–1.8)
*TP53*	mt vs. wt	0.9 (0.5–1.6)	1.3 (0.7–2.3)
*TP53 status*/*EGFR* mutation type	*TP53mt*/Exon19 vs. *TP53wt*/Exon19	0.9 (0.4–2.3)	1.2 (0.5–2.9)
	*TP53mt*/Exon21 vs. *TP53wt*/Exon19	1.2 (0.5–3.0)	1.2 (0.5–2.7)
	*TP53wt*/Exon21 vs. *TP53wt*/Exon19	1.5 (0.6–3.7)	0.9 (0.4–2.1)
Bone mets	Yes vs. No	0.9 (0.3–2.3)	1.5 (0.8–2.9)
Brain mets	Yes vs. No	3.5 (1.7–7.3) ^c^	1.8 (0.9–3.5)
Exon group	Exon21 vs. Exon19	1.4 (0.8–2.6)	0.9 (0.5–1.6)

^a^*p* ≤ 0.05; ^b^
*p* ≤ 0.01; ^c^
*p* ≤ 0.001; ^d^
*p* ≤ 0.0001. Adv dx = advanced diagnosis; ECOG PS = Eastern Cooperative Oncology Group performance status; *EGFR* = epidermal growth factor receptor; mets = metastases; NSCLC = non-small cell lung cancer; SD = standard deviation; *TP53*mt = *TP53* co-mutation; *TP53*wt = *TP53* wild-type. Cox analysis for time-to-event outcomes, and logistic regression analysis for tumor response. Other monotherapy: non-TKI monotherapies. exon 19 = exon 19 deletions, and exon 21 = exon 21L858R mutations.

## Data Availability

The data that support the findings of this study have been originated by Flatiron Health, Inc. and Foundation Medicine, Inc. These de-identified data may be made available upon request and are subject to a license agreement with Flatiron Health and Foundation Medicine; interested researchers should contact cgdb-fmi@flatiron.com and dataaccess@flatiron.com to determine licensing terms.

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
