# Peer review of "TP53* Co-Mutation Status Association with Clinical Outcomes in Patients with *EGFR*-Mutant Non-Small Cell Lung Cancer"

_cancers, 2022, doi:10.3390/cancers14246127_

Round 1
Reviewer 1 Report
In this report the authors investigate the role of TP53 co-occurring mutations with two EGFR activating mutations and how this affects cancer progression and patient survivability. Additionally, the authors take into consideration the treatment protocol for these patients. The data is sound and warrants publication but is somewhat under-described and under-analyzed. For example, what type of TP53 mutations have the authors identified co-occurring with EGFR? Are these TP53 mutations drivers, pathogenic, or just passenger. Are they homozygous or heterozygous (TP53 is a tumor suppressor so this matters). Reviewing this information is complicated by the fact that the authors do not present the raw data in the supplemental that they downloaded from the website. Although, TP53 mutations clearly have an effect, the authors should present a classification of these mutations. I suggest running the mutations through the OPEN Cravat website using the CHASM and VEST4 AI algorithms to classify the TP53 mutations. A schematic with the P53 protein coding region and where these mutations appear (e.g., lollipops) should be included. The mutations should then be labeled as driver or passenger on this schematic.
Major comments
1. In the “Materials and Methods” section, more information should be included on how the data was accessed from the website. For example, what file did you download, how was it processed, how was data extracted from the file? I attempted to go to the website but I could not find these data and could not verify the authors’ claims. For example, I could not tell if there is an answer to point 2 below.
2. What do the authors call “TP53 mutant” in Table 1? Many TP53 mutations have been identified, some pathogenic and some non-pathogenic. Further, some are driver mutations and others are not. The authors affirm that “The proportion of patients with TP53 co-mutation with exon 19 deletion or exon 21L858R mutation was 57.0% and 43.0%, respectively”. This indicates that they have stratified EGFR mutations (e.g. exon 19 or exon21) but have they stratified the TP53 mutations? This is not 100% clear from reading the text. Please make it clearer.
3. Table 4 is inconclusive since p-values are not statistically significant. Should be relegated to supplemental.
4. Figure 2A is again confusing. Which EGFR mutation is included here, or is it both? Please provide more explanation in the legend
5. Supplementary Figures 1A,B should be introduced in the main text.
6. As it is presented, Figure 3 does not add much more than Figure 2. The authors are attempting to stratify data by type of EGFR mutation but present only one statistical test that generates a p-value (between EGFR mutated and EGFR (“patients with TP53 wild-type plus EGFR exon 19 deletions, TP53 wild-type plus EGFR exon 21L858R mutations, and TP53 co-mutations plus EGFR exon 19 deletions, respectively”). In essence, this p-value represents the difference between EGFR WT and EGFR mut. The authors should make the following comparisons: EGFRexon19-TP53WT vs. EGFRexon19-TP53 mut and EGFRexon21-TP53WT vs EGFRexon21-TP53 mutant. This will tell whether there are any differences between the two types of EGFR mutations.
Minor comments
1. There is no need to partition the abstract by “background, methods, results”. Generally, papers submitted to “Cancers” as articles do not need this which is associated primarily with abstracts for clinical studies.
2. Please reference PMID: 32283832 in the intro on EGFR resistant mutations.
3. Please check grammar in detail. For example, “resistance inevitable develops” was probably supposed to say resistance inevitably develops. Numerous other examples exist throughout the paper. Check for example text fond for Fig.2 legend.
Author Response
Major comments
- In the “Materials and Methods” section, more information should be included on how the data was accessed from the website. For example, what file did you download, how was it processed, how was data extracted from the file? I attempted to go to the website but I could not find these data and could not verify the authors’ claims. For example, I could not tell if there is an answer to point 2 below.
Author Response: We thank the reviewer for the comment. Please note that the data were provided by Flatiron to Lilly under a licensed agreement and are not available on the Flatiron website. In case the reviewers would like to review the data, please see the Data Sharing Statement (Page no. 16), which states that the “The data that support the findings of this study have been originated by Flatiron Health, Inc. and Foundation Medicine, Inc. These de-identified data may be made available upon request and are subject to a license agreement with Flatiron Health and Foundation Medicine; interested researchers should contact cgdb-fmi@flatiron.com and dataaccess@flatiron.com to determine licensing terms”.
Therefore, no changes have been made to the manuscript in this regard.
- What do the authors call “TP53 mutant” in Table 1? Many TP53 mutations have been identified, some pathogenic and some non-pathogenic. Further, some are driver mutations and others are not. The authors affirm that “The proportion of patients with TP53 co-mutation with exon 19 deletion or exon 21L858R mutation was 57.0% and 43.0%, respectively”. This indicates that they have stratified EGFR mutations (e.g., exon 19 or exon21) but have they stratified the TP53 mutations? This is not 100% clear from reading the text. Please make it clearer.
Response: Thank you for the comment. Certain EGFR mutations are established driver oncogenes and commonly divided to exon 19 deletion vs. exon 21L858R mutations due to their differential responses to EGFR inhibitors in clinic. In comparison, we still don’t have treatment options specifically targeting TP53 mutations. Therefore, in the manuscript, we stratify EGFR mutations into its subtypes (exon 19 and exon 21), but do not stratify the TP53 mutation into subtypes. Throughout the manuscript (including Table 1), the TP53 mutation group includes patients with EGFR mutation (exon 19 + exon 21) and TP53 co-mutation (any subtype). We had previously acknowledged that no stratification of TP53 mutations as a limitation for this study in the discussion. “In addition, this study did not capture information on different types of TP53 co-mutation and only TP53 co-mutation at baseline were considered.” This limitation has been updated as “In addition, this study did not capture information on different types of TP53 co-mutation and only TP53 co-mutation at baseline were considered. This is because the information on TP53 mutation subtypes was not collected or available in the Flatiron database” (Page no 16)”.
- Table 4 is inconclusive since p-values are not statistically significant. Should be relegated to supplemental.
Response: Thank you for the comment. As suggested, we have moved the Table from main text to supplementary data. Now included as Supplementary Table S3 (supplementary data shared separately). Subsequent table numbers have been updated.
- Figure 2A is again confusing. Which EGFR mutation is included here, or is it both? Please provide more explanation in the legend.
Response: Thank you for the comment. A statement has been included in the footnote of figure 2 “EGFR mutation includes both exon 19 and exon 21 mutations” (Page no. 8). The text is also added as footnote for other tables and figures, wherever applicable.
- Supplementary Figures 1A,B should be introduced in the main text.
Response: We thank the reviewer for the comment, and have moved the Supplementary Figures 1A, B as Figures 3A, B (Page no. 12) per the suggestion. In addition, Supplementary Figures S2A, B are moved to the main text as Figures 4A, B (Page no. 13). Figure 3A, B from the original draft is moved to the supplementary data with updated heading (Supplementary Figure S1, B), file provided separately.
- As it is presented, Figure 3 does not add much more than Figure 2. The authors are attempting to stratify data by type of EGFR mutation but present only one statistical test that generates a p-value (between EGFR mutated and EGFR (“patients with TP53 wild-type plus EGFR exon 19 deletions, TP53 wild-type plus EGFR exon 21L858R mutations, and TP53 co-mutations plus EGFR exon 19 deletions, respectively”). In essence, this p-value represents the difference between EGFR WT and EGFR mut.
The authors should make the following comparisons: EGFRexon19-TP53WT vs. EGFRexon19-TP53 mut and EGFRexon21-TP53WT vs EGFRexon21-TP53 mutant. This will tell whether there are any differences between the two types of EGFR mutations.
Response: Thank you for the comment. Please see the response in parts below.
Part 1: The authors agree with the reviewer that Figure 2 was more important than the Figure 3. Therefore, the Figure 3 (from original manuscript) is moved to the supplementary data now (Supplementary Figure S1). However, the figure presents the 4-ways comparison between the groups included and a single p-value was generated for comparison. We have added a statement in the manuscript “Supplementary Figure S1 presents the 4-way between the survival values for the included groups.” (Page no. 11) and also added a footnote to Supplementary Figure S1 representing the same.
Comparison by TP53 mutation and exon type (2-way comparison) is included in Figure 3 and Figure 4.
Part 2: The analysis suggested by the reviewer is already presented in Supplementary Figure 3 (EGFRexon19-TP53WT vs. EGFRexon19-TP53mt) and 4 (EGFRexon21-TP53WT vs EGFRexon21-TP53mt). As stated earlier, Supplementary Figure S1, B presents the 4-ways comparison between the groups included and a single p-value was generated for comparison and exploring how the exon subtype mutation or exon wild-type may impact the outcomes in patients with TP53 mutation and TP53 wild-type-tumor. We have added a statement in the manuscript “Supplementary Figure S1 presents the 4-way between the survival values for the included groups.” (Page no. 11) and also added a footnote to Supplementary Figure S1 representing the same.
Minor comments
- There is no need to partition the abstract by “background, methods, results”. Generally, papers submitted to “Cancers” as articles do not need this which is associated primarily with abstracts for clinical studies.
Response: Thank you. We have removed the headings from the abstract (Page no. 1)
- Please reference PMID: 32283832 in the intro on EGFR resistant mutations.
Response: Thank you for the suggestion to add the reference. We have added this as references 25 (Page no. 18).
- Please check grammar in detail. For example, “resistance inevitable develops” was probably supposed to say resistance inevitably develops. Numerous other examples exist throughout the paper. Check for example text fond for Fig.2 legend.
Response: We thank the reviewer for the comment. The manuscript has been revised for grammar and performed a spell-check to remove any errors. Changes are made throughout the manuscript, as necessary.
All font sizes in footnotes and captions have been made consistent.
Reviewer 2 Report
Dear authors
I am writing about your paper entitled” TP53 Co-mutation Status Association with Clinical Outcomes in Patients with EGFR-mutant Non-small Cell Lung Cancer”. Overall scientific idea behind this paper is interesting to me. Here they should how mutation in TP63 along with EGFR effects the overall health and survival. But the main which I have is the raw data. I looked for the raw data, but its missing. Without the raw data its very hard to believe to believe the data. Author should provide the raw data. Another point is there are several papers are available online almost the same scientific idea. So how the idea behind this paper is different, author should address and modify the introduction accordingly.
- Supplementary figure 1A and 1B is missing in the main text.
- Please check the grammar.
- Should check the methodology (brief description and more clarification is needed).
Author Response
Dear authors
- I am writing about your paper entitled” TP53 Co-mutation Status Association with Clinical Outcomes in Patients with EGFR-mutant Non-small Cell Lung Cancer”. Overall scientific idea behind this paper is interesting to me. Here they should how mutation in TP63 along with EGFR effects the overall health and survival. But the main which I have is the raw data. I looked for the raw data, but its missing. Without the raw data its very hard to believe to believe the data. Author should provide the raw data. Another point is there are several papers are available online almost the same scientific idea. So how the idea behind this paper is different, author should address and modify the introduction accordingly.
Response: We thank the reviewer for the comment. The data were received from Flatiron as a part of a licensed agreement between Flatiron and Lilly and are not available on the website. In case the reviewers would like to review the data, please see the Data Sharing Statement (Page no. 17), which states that the “The data that support the findings of this study have been originated by Flatiron Health, Inc. and Foundation Medicine, Inc. These de-identified data may be made available upon request and are subject to a license agreement with Flatiron Health and Foundation Medicine; interested researchers should contact cgdb-fmi@flatiron.com and dataaccess@flatiron.com to determine licensing terms”.
As our data was based on Flatiron clinical information and Foundation Medicine genetic profiling information, our analysis was limited to the Foundation Medicine gene panel. Currently, TP63 is not tested in Foundation Medicine panel, therefore, there is no data regarding TP63.
Regarding the differentiation from other studies, we have modified the statement in the introduction “To provide further body of evidence by building on previous work (30-38), this study examined the impact of TP53 co-mutations on treatment outcomes in a cohort with EGFR-mutated aNSCLC using a nationally representative electronic medical records (EMR)-derived database linked with next-generation sequencing data.” (Page no. 2)
- Supplementary figure 1A and 1B is missing in the main text.
Response: We thank the reviewer for the comment and have moved the Supplementary Figures 1A, B as Figures 3A, B (Page no. 12) per the suggestion. In addition, Supplementary Figures S2A, B are moved to the main text as Figures 4A, B (Page No. 13). Figure 3A, B from the original draft is moved to the supplementary data (Supplementary Figure S1, B), file provided separately.
- Please check the grammar.
Response: We thank the reviewer for the comment. The manuscript has been revised for grammar and performed a spell-check to remove any errors. Changes are made throughout the manuscript, as necessary.
- Should check the methodology (brief description and more clarification is needed).
Response: We thank the reviewer for the comment. The changes have been made in the methodology for clarification as needed (Page no. 2 to 4).
Reviewer 3 Report
This manuscript describes a relatively large clinicogenomic study describing TP53 co-mutation with EGFR mutated in NSCLC is associated with poor prognosis. The study focuses on treatment outcomes and survival as way to identify subsets of EGFR mutation and p53 co-mutation where treatments can be improved. Overall, the study was well-written and informative for researchers in the field. Importantly, the authors identified and discussed the strengths and limitations of their study.
Author Response
This manuscript describes a relatively large clinicogenomic study describing TP53 co-mutation with EGFR mutated in NSCLC is associated with poor prognosis. The study focuses on treatment outcomes and survival as way to identify subsets of EGFR mutation and p53 co-mutation where treatments can be improved. Overall, the study was well-written and informative for researchers in the field. Importantly, the authors identified and discussed the strengths and limitations of their study.
Response: Dear reviewer, we thank you for the words of appreciation regarding the design, analysis, writing, and the methods of presentation.
Reviewer 4 Report
TP53 Co-mutation Status Association with Clinical Outcomes in Patients with EGFR-mutant Non-small Cell Lung Cancer
Xiuning Le, et. al., MDPI Cancers
Summary: Le et. al. present a retrospective to assess the severity and outcomes of having mutant p53 in additional to clinically treated mutant EGFR.
The analyses presented are sufficiently strong, and support the claims made by the authors.
I recommend publishing this study.
Comments:
-
The study is well-written, and its limitations are enumerated.
-
It is largely agreed upon in the field, and proven by mouse models of the disease - including mouse models of conditional p53 loss - that acquired p53 mutations during a tumor’s progression result in worse disease outcomes. These retrospective analyses of deidentified patient samples add an additional layer of proof to that concept.
-
The authors mention an apparent contradiction of mutant p53 co-occurrence in the introduction: this could be further elaborated on and presented with putative reasons as to why there are two schools of thought regarding this.
Author Response
Summary: Le et. al. present a retrospective to assess the severity and outcomes of having mutant p53 in additional to clinically treated mutant EGFR.
The analyses presented are sufficiently strong, and support the claims made by the authors.
I recommend publishing this study.
Comments:
- The study is well-written, and its limitations are enumerated. It is largely agreed upon in the field, and proven by mouse models of the disease - including mouse models of conditional p53 loss - that acquired p53 mutations during a tumor’s progression result in worse disease outcomes. These retrospective analyses of deidentified patient samples add an additional layer of proof to that concept.
Response: Dear reviewer, we thank you for the words of appreciation regarding the design, analysis, writing, and the methods of presentation.
- The authors mention an apparent contradiction of mutant p53 co-occurrence in the introduction: this could be further elaborated on and presented with putative reasons as to why there are two schools of thought regarding this.
Response: We thank the reviewer for the comment. The introduction part in the manuscript has been revised per suggestion.
“Previous studies examining the impact of co-mutations in EGFR-mutated aNSCLC treated with EGFR-TKIs demonstrated that the presence of TP53 co-mutation might be a marker of poor prognosis (28, 30-33). However, other studies found no association between presence of TP53 co-mutations and clinical outcomes (34-38). The conflicting results were likely due to small cohort sizes, for example n=37 in one study (34), or as well as due to differences in statistical analyses (e.g., univariate vs multivariable analyses accounting for known prognostic factors such as age, baseline metastases, and ECOG performance status) (35). In early disease stage lung cancers (stage I-III), role of TP53 mutation is more controversial (36, 37). To provide body of evidence by building on previous work (30-38), this study examined the impact of TP53 co-mutations on treatment outcomes in a cohort with EGFR-mutated aNSCLC using a nationally representative electronic medical records (EMR)-derived database linked with next-generation sequencing data. Subgroup analyses are reported based on EGFR mutation subtype and EGFR-TKI generation.”
Round 2
Reviewer 1 Report
The authors have made significant changes to this revision. This reviewer is satisfied.